# Femtosecond switching of strong light-matter interactions in microcavities with two-dimensional semiconductors

Armando Genco [1,11], Charalambos Louca [1,2,11], Cristina Cruciano[1], Kok Wee Song [3,4], Chiara Trovatello [1,5], Giuseppe Di Blasio[1], Giacomo Sansone [6], Sam A. Randerson [7], Peter Claronino[7], Kyriacos Georgiou [8], Rahul Jayaprakash [7], Kenji Watanabe [9], Takashi Taniguchi [9], David G. Lidzey [7], Oleksandr Kyriienko [3,7], Stefano Dal Conte [1], Alexander I. Tartakovskii [7] ✉ & Giulio Cerullo [1,10] ✉

Ultrafast all-optical logic devices based on nonlinear light-matter interactions hold the promise to overcome the speed limitations of conventional electronic devices. Strong coupling of excitons and photons inside an optical resonator enhances such interactions and generates new polariton states which give access to unique nonlinear phenomena, such as Bose-Einstein condensation, used for all-optical ultrafast polariton transistors. However, to reach the threshold for condensation high quality factors and high pulse energies are required. Here we demonstrate all-optical switching exploiting the ultrafast transition from the strong to the weak coupling regime in low-Q microcavities embedding bilayers of transition metal dichalcogenides with high optical nonlinearities and fast exciton relaxation times. We observe a collapse of polariton gaps as large as 55 meV, and their revival, lowering the threshold for optical switching below 4 pJ per pulse, while retaining ultrahigh switching frequencies. As an additional degree of freedom, the switching can be triggered pumping either the intra- or the interlayer excitons of the bilayers at different wavelengths, speeding up the polariton dynamics, owing to unique interspecies excitonic interactions. Our approach will enable the development of compact ultrafast all-optical logical circuits and neural networks, showcasing a new platform for polaritonic information processing based on manipulating the light-matter coupling.

All-optical switches based on nonlinear optical materials have been extensively investigated to overcome the speed limitations of electronic circuits, thanks to their potential to work at much higher frequencies owing to the inherently fast light-matter interactions underlying their operation[1]. Demonstrations of ultrafast all-optical logic gates have been achieved in a plethora of solid state platforms, exploiting optical nonlinearities ($\chi^{(2)}$ and $\chi^{(3)}$)[2,3] and saturable absorption[4]. Notable examples employed microring resonators[5], plasmonic nanostructures[6], photonic crystals[7], metasurfaces[8], 2D materials[9] or even single molecules[10]. Such devices showcased switching times down to tens of femtoseconds, but usually at the expense of the switching energy or the on/off contrast[11,12]. More recently, an optimal combination of femtosecond switching times and femtojoule operating energies has been obtained in an all-optical

nonlinear device based on a lithium niobate waveguide[13], but with millimeter-scale lengths hindering the production of densely on-chip integrated circuits. Therefore, achieving high performances in all-optical switches is still an open challenge, with the future perspective of integrating them in a compact photonic processor.

Excitons in semiconducting materials embedded in optical resonators can be used for all-optical switching, as they show a highly nonlinear response when they are in strong coupling (SC) with resonant photons confined in the structure. In such a regime, the rate of coherent energy transfer between the energy-degenerate excitons and photons is higher than the loss rate, and new hybrid light-matter quasiparticles arise, called polaritons[14]. The energy splitting of the polariton states (Rabi splitting) is a direct measure of their coupling strength, which is proportional to the quality factor (Q factor) of the resonator and to the exciton absorption cross section.

Harnessing polaritonic nonlinear interactions is of key importance for a broad range of phenomena and applications, such as lasing[15], optical parametric amplification[16], Bose-Einstein condensation (BEC)[17], or for quantum effects (polariton blockade)[18,19]. A combination of blueshift of the polariton states and gain in photoluminescence intensity occurring above the threshold of BEC has been used as operational principle of all-optical polariton logic devices and neural networks[20–22]. Ultrafast optical switching (with switching times ≤1 ps) relying on BEC has been demonstrated even at room temperature[23–25], but typically using high Q factor architectures and high pulse energies to reach the condensation threshold, i.e. from tens to hundreds pJ per pulse[26,27], although the energy/pulse thresholds for polariton nonlinearities have been recently brought down from ~10 nJ[28] to ~1 pJ[29].

Alternative strategies for all-optical polariton switching have been also shown[30–33], used, for example, for optical spin switches[34,35]. A promising approach to achieve high performances in polariton switching relies on the modulation of the light-matter coupling strength, leveraging on the exciton absorption saturation, which produces a spectral shift of the polariton states, acting as a gate for the light transmitted/reflected by the device. Varying strongly the coupling strength would eventually lead to a complete transition from strong to weak coupling regime or viceversa[36]. SC can be switched off in strongly coupled optical microcavities comprising GaAs quantum wells through optical saturation of excitons[36,37] or via electrically-tuned charge build-up[38]. Alternatively, it can be switched on by optically induced absorption, i.e. for inter sub-band transitions[39]. However, achieving complete on/off switching cycle below 1 ps acting on the coupling strength in these material platforms is not possible due to the long excitons and excited carriers lifetime[40].

Atomically thin transition metal dichalcogenides (TMDs) are promising nonlinear optical materials[41], where excitons remain stable up to room temperature due to large binding energies and oscillator strength[42,43]. Owing to these properties, TMDs can easily enter the SC regime, when integrated in optical resonators[44,45]. Recent studies of TMD polaritons aim to maximize nonlinear interactions going beyond the use of 1s neutral excitons, exploring higher Rydberg excitonic states, charged excitonic complexes, moiré or dipolar excitons[46–50]. Moreover, the fast dynamics of TMD excitons[51] makes these materials very promising for ultrafast logic gates. Optically pumping TMD monolayers coupled to optical resonators with ultrashort laser pulses can modulate[52] or completely quench the Rabi splitting[53] increasing the pump fluence, owing to strong exciton nonlinear interactions. However, a time-resolved study of the complete strong-to-weak coupling transition in microcavities with atomically thin TMDs and the demonstration of its use for high performance all-optical switching have never been shown.

Here we use MoS$_2$ bilayers embedded in low-Q factor microcavities to produce an ultrafast collapse and revival of the SC regime using very low pulse energies ( <4 pJ). Compared to monolayers, bilayers offer a unique combination of crucial properties to obtain such effect, such as (i) ultrafast and efficient exciton relaxation, (ii)

strong nonlinearities, i.e. Coulomb dipole-dipole interactions and phase space filling, enhanced by the reduced dielectric screening[54], (iii) hybridized interlayer excitons with high oscillator strength[55], leading to distinctive interspecies intra-interlayer exciton interactions[54]. We employ femtosecond transient reflectivity (TR) spectroscopy to demonstrate the ultrafast switching of the SC regime in compact devices at both cryogenic and room temperature (RT). We show the full tunability of this process through different degrees of freedom, such as pump wavelength, pulse power and cavity detuning. The SC switching leads to a strong modulation of the polariton peaks splitting, reducing the initial energy separation from 42 meV to less than their linewidth, making them indistinguishable from a single peak. The Rabi splitting modulation is further enhanced by placing a stack of two bilayers separated by hBN in the cavity, going from 55 meV to a complete collapse, resulting in an effective extinction ratio of about 7.5 dB working in reflection. We further demonstrate an on/off SC switching frequency as high as 250 GHz, which can be extended up to 1 THz.

## Results

### Static and dynamic optical behavior of MoS$_2$ bilayers

The TMD structures used in our work are made of monolayers (MLs) and bilayers (BLs) of MoS$_2$ encapsulated in hBN, placed on distributed Bragg reflectors (DBRs), for the subsequent fabrication of optical microcavities. Unless specified, all the spectroscopy experiments in this work are performed at T = 8K. Figure 1a shows the Reflectance Contrast (RC) spectra of ML and BL MoS$_2$ outside the cavity. The absorption of the intralayer A exciton in the BL ($X_{A-BL}$) is higher than in the ML, due to presence of the additional layer. In the BL a new excitonic resonance appears at ≈2 eV, which is attributed to dipolar hybridized interlayer excitons (hIX) with a high oscillator strength, resulting from the coherent tunneling of holes between the valence bands of the two layers (Fig. 1a inset)[55,56].

We study the ultrafast response of MoS$_2$ excitons by ultrafast TR micro-spectroscopy. We deliver two collinear pulses, a narrow-band pump and a broad-band probe, focused on the sample using a microscope objective (Fig. 1b). We then vary the delay time $\tau$ between them and monitor the changes in the broadband reflectivity spectrum of the probe (see Methods for experimental details). Figure 1c shows the differential reflectivity ($\Delta R/R$) map measured as a function of the probe photon energy and pump-probe delay for a BL MoS$_2$, tuning the energy of the pump pulses at ≈1.94 eV, in resonance with the $X_{A-BL}$.

Generally, the shape of the TR spectra in TMD MLs is a result of multiple effects, such as optical saturation (photo-bleaching), line broadening and spectral shift of the exciton peaks, leading to positive and negative TR signals around the exciton energies[57]. To show more clearly the temporal evolution of the exciton features in our system, we perform an analysis of the transient $\Delta R/R$ response based on the Transfer Matrix Method (TMM) to extract the time-dependent RC spectra of the material (see Methods for details)[58]. Figure 1d shows the MoS$_2$ BL dynamic RC spectrum at a delay of 0.3 ps (purple curve), compared to the one before the pump pulse ( − 0.3 ps, orange curve). We fit the dynamic RC with three Lorentzians (solid lines in Fig. 1d) to extract the time-varying intensity and energy shift of each excitonic mode. After excitation, the $X_{A-BL}$ peak is quenched and slightly blue-shifted. The small shoulder at 1.91 eV appearing at positive delays is instead related to a photo-induced absorption of the trion ($X^{*}_{A-BL}$)[59–61]. Since the excitation pulses are in resonance with $X_{A-BL}$ (shaded yellow area in Fig. 1d), at lower energies compared to hIX, we would expect negligible optical saturation of the latter if the two excitonic species were totally uncoupled. On the contrary, we observe a photo-bleaching of the hIX absorption, although less intense than in the $X_{A-BL}$ case. This indicates their hybridization with intralayer excitons due to the coherent hole tunneling between the valence bands of the two layers and to the fermionic interactions between holes of $X_{A-BL}$ and hIX sharing the same valence band (see inset of Fig. 1a)[54].

Tracking the RC peak intensity as a function of the delay time, we can extract the ultrafast dynamics of the excitonic species (Fig. 1e). The transient behavior of exciton energies and linewidths is shown in Supplementary Note S1. For both $X_{A-BL}$ and hIX, the exciton population rises instantaneously (within the $\approx 100$ fs temporal resolution of our setup), then decays exponentially, with about 50% of the initial population already relaxed within 2 ps (a comparison with MoS$_2$ ML exciton dynamics is reported in Supplementary Note S2). The fast exciton decay time is similar between $X_{A-BL}$ and hIX.

The ultrafast nonlinear optical response of mono and few-layers TMDs has been studied extensively in the past[62–64]. Transient exciton line shifts in TMDs are usually ascribed to Coulomb interactions at short time scales (few ps)[58,65], or bandgap renormalization[66], and to transient heating effects[67] at longer times (from tens to hundreds of ps). Exciting TMD monolayers close or below the exciton energy also leads to strong and instantaneous (within the pump pulse duration) line shifts due to the optical Stark effect[68,69]. High exciton densities in TMDs lead to optical saturation, due to phase-space filling (i.e. Pauli blocking)[67], and line broadening caused by excitation-induced dephasing[57]. Tracking the time-dependent exciton saturation in ultrafast pump-probe experiments allows monitoring the exciton population dynamics.

In MoS$_2$ BLs we observe a bi-exponential population decay with a fast and a slow component. While in MLs the fast decay is usually attributed to radiative and non-radiative relaxation processes of bright excitons[70], in BLs it is more probably related to electron-phonon intervalley scattering processes from the K points to the lowest energy point of the Brillouin zone[71–73]. The slow decay component can be related to phonon-assisted recombination from dark states[70] or defect-mediated non-radiative recombination[74].

## Femtosecond switching of the strong coupling regime

We exploit the highly nonlinear exciton interactions in MoS$_2$ BL to drastically modify the light-matter coupling strength in microcavities on ultrafast time-scales. The microcavity samples are fabricated by covering the hBN-encapsulated MoS$_2$ heterostructures placed on DBRs with a transparent polymeric spacer (polymethylmethacrylate, PMMA) and a top silver (Ag) mirror, as illustrated in Fig. 2a. We perform k-space (Fourier) spectroscopy to image the angular dispersion of the monolithic cavity embedding the MoS$_2$ BL (Fig. 2b). Two distinct anticrossings appear when the cavity mode is in resonance with $X_{A-BL}$ and hIX energies, a clear signature of the SC regime, resulting in upper, middle and lower polariton branches (UPB, MPB, LPB). Fitting the dispersion with a three coupled oscillators model, we extract Rabi splittings of $\Omega_{A_{BL}}$ = 42 meV and $\Omega_{hIX}$ = 23 meV for $X_{A-BL}$ and hIX,

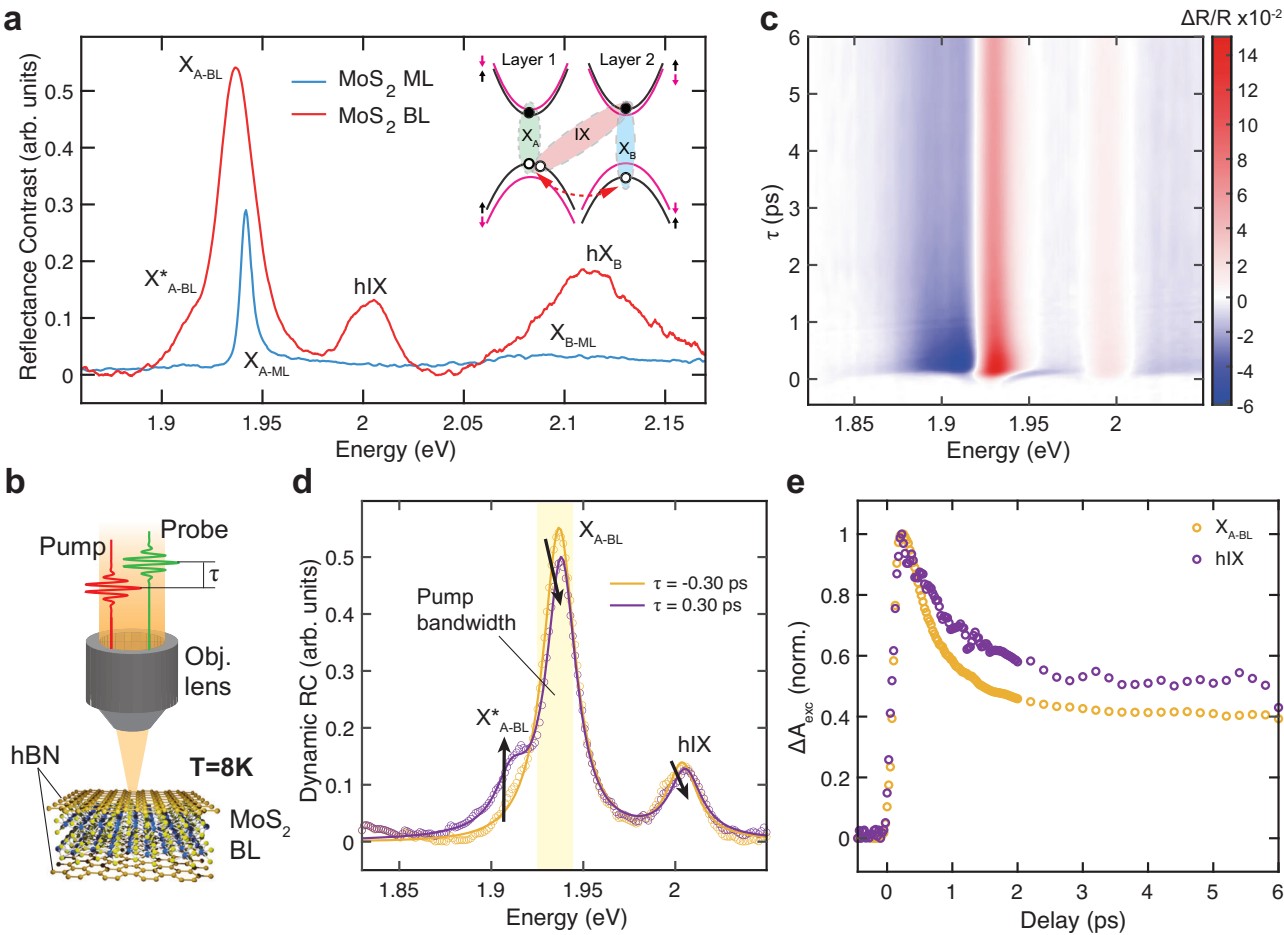

**Fig. 1 | Optical characterization of the BL MoS$_2$. a** Static RC spectra of a ML and BL MoS$_2$ encapsulated in hBN. RC = $(R_{sub} - R_{TMD})/R_{sub}$, where $R_{TMD}$ is the reflectance of the sample, while $R_{sub}$ is taken on the substrate. Inset: sketch of the band diagram in MoS$_2$ BL; the dashed red line indicates the coherent tunneling of holes. **b** Sketch of the MoS$_2$ BL encapsulated in hBN (out of scale) measured by pump-probe microspectroscopy. **c** Transient differential reflectivity map as a function of delay time $\tau$ and probe photon energy measured for MoS$_2$ BL. **d** Dynamic RC of the MoS$_2$ BL at negative (before pulsed excitation) and positive (after pulsed excitation) delay times, extracted from the differential reflectivity map in **c**. Solid curves show the Lorentzian fit of the dynamic RC. Black arrows show optical saturation and energy shift of the $X_{A-BL}$ and hIX transitions, and the photo-induced absorption of the $X^*_{A-BL}$ trion. The shaded yellow area displays the energy and bandwidth of the pump pulses. **e** Normalized exciton peak amplitude variation ($\Delta A_{exc}$) of $X_{A-BL}$ and hIX, extracted from the dynamic RC at different time delays.

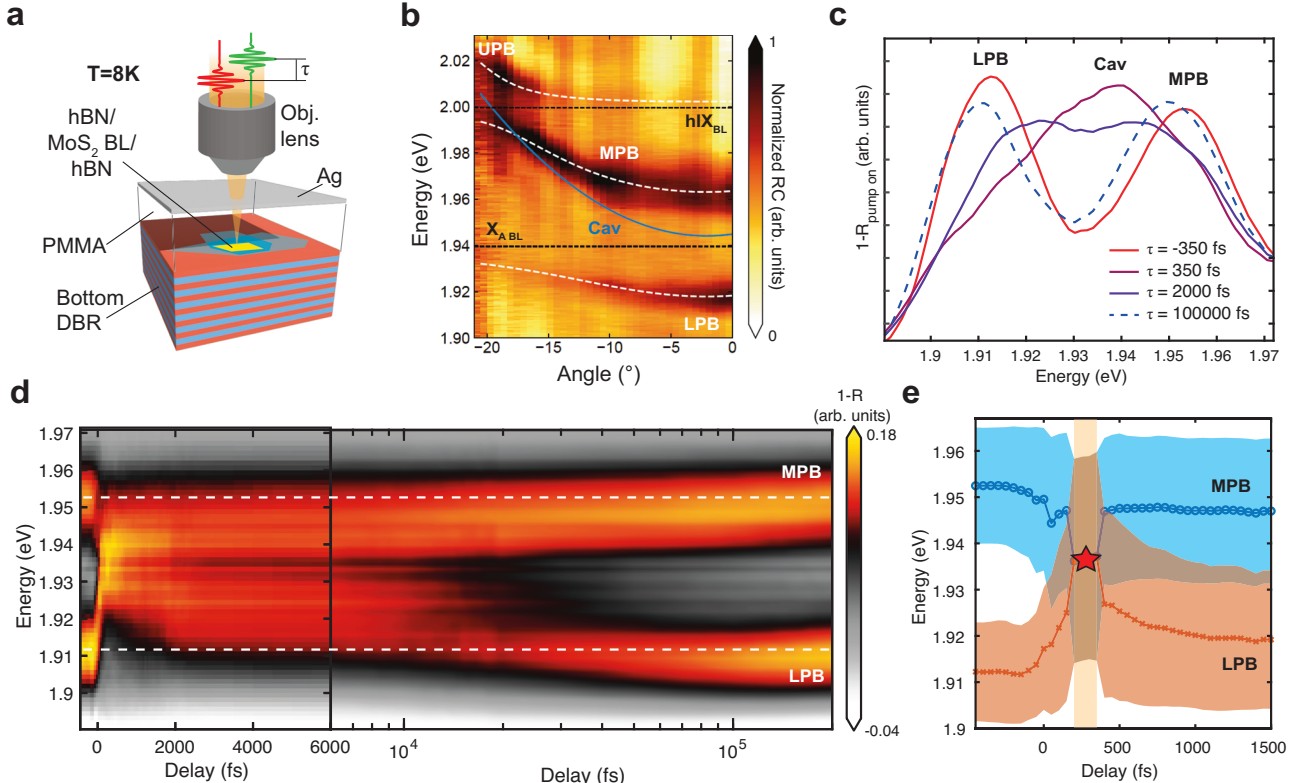

**Fig. 2 | Ultrafast switching of strong coupling in a MoS₂ BL microcavity. a** Sketch of the MoS₂ BL microcavity structure measured by pump-probe spectro-microscopy. **b** Color map of the angle-resolved RC spectra of a microcavity embedding a BL MoS₂ in strong coupling regime, showing two distinct anticrossings around the $X_{A-BL}$ and hIX energies (black dashed lines) respectively. The coupled oscillators model fit (white dashed lines) and the cavity mode dispersion (blue line) are shown in overlay. **c** 1-R spectra of the BL microcavity taken at different pump-probe delays, pumping the system at 1.94 eV with 3.75 pJ. Immediately after excitation, the polariton branches collapse in a central weakly coupled cavity mode. **d** Color map

of the 1-R spectra of the BL microcavity as a function of the pump-probe delay showing the ultrafast collapse and later revival of the MPB and LPB (white dashed lines). **e** Results of Gaussian fits of the polariton/cavity modes dynamic spectra extracted from Fig. 2d. The blue (orange) trace refers to the MPB (LPB) peak energy, while the shaded areas depict the linewidth of the modes (Full Width Half Maximum, FWHM). Under the shaded yellow area only a weakly coupled cavity mode can be fitted, with the red star highlighting the crossing region between strong and weak coupling regimes.

respectively. We also fabricated a microcavity with a similar structure embedding a MoS₂ ML, which shows an anticrossing between the cavity mode and intralayer excitons with a Rabi splitting of $\Omega_{A_{ML}} = 28$ meV. The latter is reduced compared to the BL cavity due to the lower absorption (see Supplementary Note S3 for the static analysis of the ML cavity).

We use ultrafast TR spectroscopy to excite the MoS₂ BL-based microcavities with narrowband ultrashort pulses tuned at the energy of $X_{A-BL}$. To better visualize the dynamic behavior of polariton spectrum, we plot directly the reflectance (1-R) spectra measured on the cavity as a function of delay time and probe photon energy (Fig. 2c, d), while we include the TR data of the same measurement in Supplementary Note S4. Considering that in the spectral region of interest the reflectance of the cavity without the TMD is close to 1, plotting 1-R as a function of time is equivalent to showing the dynamic RC. We focus our analysis on incidence angles close to normal, on the anticrossing between the cavity mode and the $X_{A-BL}$, resulting in the MPB and LPB. In a stark contrast to the out-of-cavity experiments, we do not observe a direct reduction of the exciton absorption in this measurement, but we monitor it indirectly through huge shifts of the polariton states. At negative delays, MPB and LPB are clearly separated, located at 1.953 eV and 1.911 eV respectively. When the pump and probe pulses are synchronous, the two polariton peaks collapse symmetrically in one broad central peak at ≈ 1.94 eV (purple line in Fig. 2c). Already after 2 ps, the two polariton branches start to reappear, while after 100 ps they have almost completely recovered. The 1-R map as a function of the time

delay (Fig. 2d) shows more clearly the complete collapse and revival of polaritons, which can be only explained as a reversible transition from the strong to the weak coupling regime. In our system, the collapse of SC is mostly related to a large density of uncoupled excitons which saturates the optical transition. The SC recovery is consequent to the relaxation of such excitons, leading to a regaining of oscillator strength. In fact, we note that the SC recovery follows well the dynamic absorption of the excitonic species measured outside the cavity (Fig. 1e and Supplementary Note S5), being a direct consequence of density-dependent optical saturation of excitons. The two polariton branches show different recovery times depending on their Hopfield coefficients, and in particular on their photonic component. In fact, a polariton branch with a larger photonic character will be closer in energy to the weakly coupled cavity mode, leading to a faster recovery. Therefore, a positive detuning benefits the MPB recovery over the LPB one, as shown in Fig. 2d, while the opposite happens for negative detunings (see Fig. S13).

We performed a quantitative analysis of the ultrafast behavior of MoS₂ BL polaritons by fitting the experimental 1-R peaks with Gaussian functions. The results are shown in Fig. 2e, where the extracted peak energies and linewidths are plotted against the time delay up to 1.5 ps. Within few hundreds of femtoseconds from the zero-delay, the LPB shows a blueshift of about 27 meV, while the MPB redshifts by about 14 meV, merging in a single peak at about 250 fs. Such huge shifts cannot be explained just taking into account the bare exciton energy variations, which are in the order of only a few meVs (Fig. 1d). When the

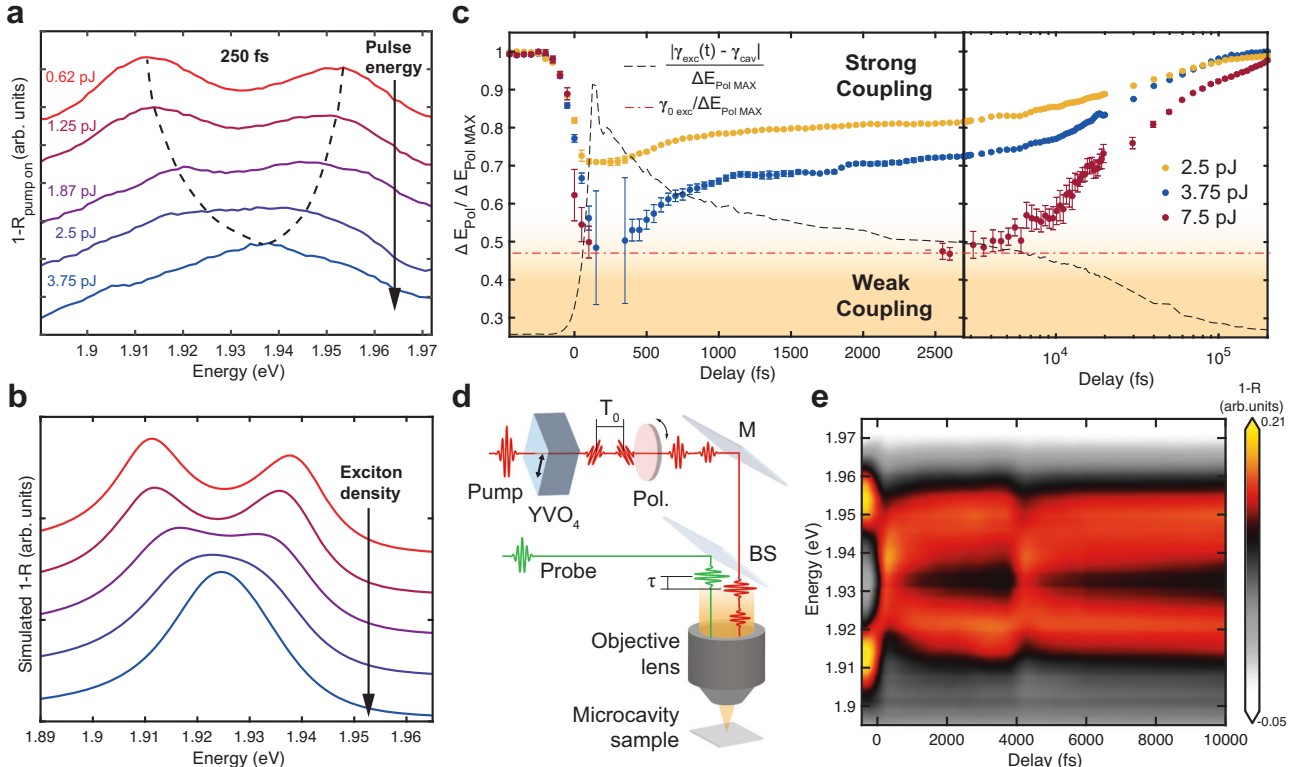

**Fig. 3 | Control of strong coupling switching. a** BL cavity 1-R spectra taken at a delay time of 250 fs, pumping the system at increasingly higher pulse energies. **b** Simulated cavity spectra for increasing exciton-polaritons densities in the MoS$_2$ BL. **c** MPB-LPB energy difference as a function of delay time for ultrafast SC switching experiments in BL cavities at different pump pulse energies, normalized to the value before excitation. Error bars are drawn from the confidence interval of the double peak fits. The dashed black curve shows the trend of the exciton-cavity linewidth difference, while the red dashed-dotted line shows the linewidth of the unperturbed exciton, both normalized by the polariton splitting value before excitation. The weak coupling time window duration can be tuned by changing the excitation pulse energy. **d** Sketch of the experimental configuration used to produce delayed double pump pulses. M mirror, BS beam splitter, Pol polarizer. **e** Color map of the 1-R spectra of the BL microcavity as a function of the delay time, excited by double pump pulses delayed by ≈4 ps.

energy separation of the polariton states is lower than the linewidth of the cavity mode or the exciton, the anticrossing is not visible anymore and the system falls into the weak coupling regime (red star in Fig. 2e). Already after ≈ 500 fs, the SC is recovered. The Q factor of our cavity is about 190, leading to a photon lifetime of ~ 65fs, being much faster than the observed recovery dynamics. This suggests that such behavior is dominated by incoherent excitonic processes. We note that in the weak coupling, the cavity mode is strongly broadened by the background absorption of the excitons, already broad due to excitation-induced dephasing[57]. Such broadening also affects the polariton peaks after the collapse, as shown in the color bars of Fig. 2e, which become more discernible only after 2 ps. The polariton linewidths narrow down even more after 10 ps, when the effects of excitation induced dephasing fade away, as shown in Supplementary Note S11. To a first approximation, we can consider that the strong to weak coupling full transition is reached when the Rabi splitting is equal or below the unperturbed exciton linewidth (the FWHM of $X_{A-BL}$, $\gamma_{0exc}$, is ~ 20 meV in static conditions). A more precise definition of strong to weak coupling threshold implies that the energy exchange between cavity and exciton resonances is larger than the difference between the loss rates[75,76]. On the other hand, considering in our case the exciton line broadening caused by excitation-induced dephasing, this becomes a less stringent criterion, as discussed later in this section (Fig. 3c).

Leveraging on the large binding energy of excitons in TMDs, we fabricated an additional BL MoS$_2$ microcavity in SC regime at RT. We performed a full SC switching also in this device at ambient conditions, shifting the LP by about 20 meV using a pump pulse energy of ~ 1.8 pJ (see Supplementary Note S11).

Finally, we observed a similar SC collapse also in the microcavity embedding a ML of MoS$_2$, but in that case the longer exciton lifetimes led to a much slower SC recovery, while the smaller Rabi splitting worsened the on/off contrast, i.e. the signal intensity ratio between the 1-R spectra of the cavity in the unperturbed SC and weak coupling conditions respectively (see Supplementary Note S6). The switching contrast is influenced by a number of factors. The most important ones are the visibility of the polariton modes, controlled by the detuning, the maximum achievable dynamic energy shift, directly proportional to the Rabi splitting, and the exciton and polariton linewidth broadening. The latter can significantly worsen the switching contrast and is also dependent on the exciton density and the pump fluence. Reducing the static and dynamic exciton and polariton broadening or enlarging the Rabi splitting will increase the on/off contrast.

The pump pulse energy plays a major role in the SC switching dynamics, as shown in Fig. 3a where MoS$_2$ BL cavity spectra taken at a delay of 250 fs for different excitation pulse energies demonstrate the gradual quenching of the Rabi splitting. The SC collapse in TMD cavities is a direct consequence of exciton nonlinear interactions, which scale proportionally to their density[54]. We demonstrate this effect by carrying out theoretical simulations of the cavity 1-R spectra using the TMM (Fig. 3b), employing the MoS$_2$ BL optical constants calculated from the exciton nonlinear absorption as a function of the density (see Supplementary Note S7). The match between experiments and simulations proves that the main cause behind the observed femtosecond switching of the SC regime is the optical saturation and broadening of MoS$_2$ BL excitons at high excitation densities, which recovers very rapidly due to the fast radiative and non-radiative exciton relaxation mechanisms in this system.

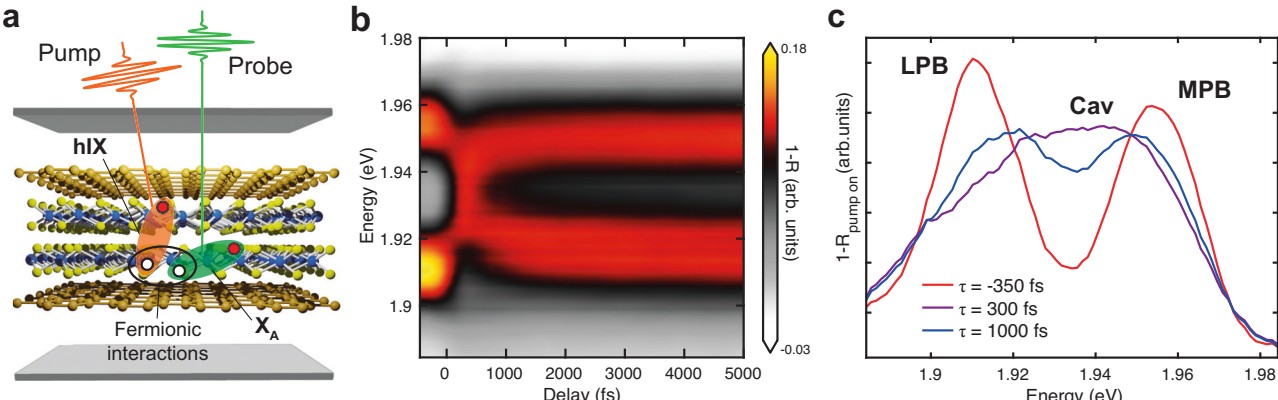

**Fig. 4 | Ultrafast strong coupling switching by interspecies interactions.**
**a** Sketch of the MoS₂ BL cavity measured in pump-probe, exciting the hIX spectral region and probing the polariton states formed around $X_{A-BL}$. **b** Color map of the 1-R spectra of the BL microcavity as a function of the delay time, exciting the hIX and probing the MPB-LPB spectral region. **c** 1-R spectra of the BL microcavity pumping the hIX, taken at different pump-probe delay times. Already after 1 ps, the polariton peaks are clearly recovered.

Figure 3c reports the MPB-LPB energy difference against the time delay, normalized with respect to its value before excitation. In this figure, the blue dots are related to the experiment reported in Fig. 2, performed at 3.75 pJ (pump fluence: 212 $\mu$J cm⁻²), and show the recovery of SC occurring on two different time-scales, a fast one within 1 ps and a slow one which is concluded after ≈ 100 ps. We ascribe those two recovery steps to the population decay dynamics of the bare excitons (see Fig. 1e and Supplementary Note S5). The red dashed horizontal line in Fig. 3c represents the threshold when the polariton splitting is smaller than the unperturbed exciton linewidth ($\gamma_{0exc}$), while the black dashed curve shows the time-dependent normalized difference between exciton and cavity linewidths. While the cavity linewidth remains approximately the same in all the experiments ( ~ 10 meV), the exciton linewidth changes with fluence and time because of excitation-induced dephasing. We extract the transient $X_{A-BL}$ linewidth, $\gamma_{exc}(t)$, analysing the time-dependent reflectivity of the out-of-cavity sample, excited with a pump fluence comparable to the ones used in the cavity experiments (see Supplementary Note S5). Using the exciton-cavity linewidth difference to set the threshold for SC, the switching is not as sharp, but it still occurs in a sub-picosecond time window, between ~ 50 fs and ~ 700 fs, pumping with 3.75 pJ. On the other hand, using such a definition for the strong to weak coupling transition, the pulse energy to induce the SC collapse will decrease. At the SC switching pump energy threshold, we estimated a peak polariton density of about 10⁵$\mu$m⁻². Increasing or decreasing by few picojoules the excitation energy, we can extend the temporal window of weak coupling regime (7.5 pJ, red dots in Fig. 3c) or suppress the transition (2.5 pJ, yellow dots in Fig. 3c).

We provide a theoretical explanation on the energy dependent dynamic SC switching upon direct excitation of the intralayer excitons, considering several possible contributions to the dynamics. We note that the ≈ 1 ps timescale for the fast recovery coincides with the bare $X_{A-BL}$ exciton fast decay. However, this short timescale cannot account for the later slower recovery ( ~ 100 ps) of the Rabi splitting, indicating that a significant exciton population remains in the sample and non-linear phase space filling impacts the spectrum. One plausible explanation is the existence of long-lived dark excitonic states at lower energies[77,78]. In this case photoexcited excitons can be transferred to such states which interact with light weakly, forming a reservoir that contributes to the nonlinear phase space filling. In a bilayer MoS₂, low-energy states are represented by spin-forbidden states due to spin-orbit coupling, or momentum-forbidden states[77] due to indirect bandgap[78]. These states possess a very long lifetime and can explain the TR dynamics. The corresponding model is summarized in the

Methods, while we provide more details about the simulated polariton dynamics in Supplementary Note S8.

Leveraging on the ultrafast recovery times of SC in our samples, we demonstrate the possibility to modulate light-matter interactions at very high frequencies, illuminating the cavity with two subsequent pump pulses at 1.94 eV delayed by only ≈ 4 ps. To produce such pulse pair, we used a birefringent YVO₄ crystal with optical axis rotated by 45° with respect to the polarization of the incoming pump pulse, followed by a linear polarizer (see Fig. 3d and Methods for more details). The first pulse energy was tuned to be slightly lower than in the experiment of Fig. 2 in order to get a faster SC recovery, while the second pulse energy was adjusted to take into account the residual exciton population after the first pulse. The resulting transient 1-R map shows two reversible on/off cycles (Fig. 3e), proving a very fast switching frequency of ≈ 250 GHz. We note that this value was limited by technical constraints (the fixed delay between the pulse pair determined by the thickness of the available YVO₄ crystal), while the theoretical limit is given by the recovery time of the SC.

## Ultrafast SC switching by interspecies interactions

We exploit the interspecies exciton interactions specific of MoS₂ BL to generate optical saturation of $X_{A-BL}$ acting on the hIX, exciting selectively the latter and probing the quenching of the Rabi splitting on $X_{A-BL}$ (Fig. 4a). This process relies on nonlinear fermionic interactions (i.e. involving a single charge carrier constituting the exciton) between the two excitonic species: the $X_{A-BL}$ valence band is shared with hIX, therefore exciting the latter causes optical quenching of the former, due to Pauli blocking of holes for $X_{A-BL}$[54]. Figure 4b shows the transient 1-R map of the MPB-LPB, pumping the hIX of the BL: the femtosecond switching of SC regime occurs very clearly also in this case. Comparing this result with the previous case of resonant $X_{A-BL}$ pumping (Fig. 2c), the fast SC recovery is even more distinct, with the two polariton peaks being clearly visible and well separated already after 1 ps, as shown in Fig. 4c.

The dynamics of the nonlinear response when pumping in resonance with the hIX is also consistent with the developed model based on the nonlinear saturation and phase space filling from the long-lived states (see Methods). In this case, we considered similar lifetimes for hIX compared to $X_{A-BL}$, but we assumed the rate for transferring the pumped hIX to the long-lived reservoir states contributing to the phase space filling to be smaller than in the $X_{A-BL}$ case. This may be understood as the result of the hIX's wavefunction spreading in the out-of-plane direction. This implies that hIX is less 2D than $X_{A-BL}$, leading to a weaker scattering effect with disorder and to a smaller

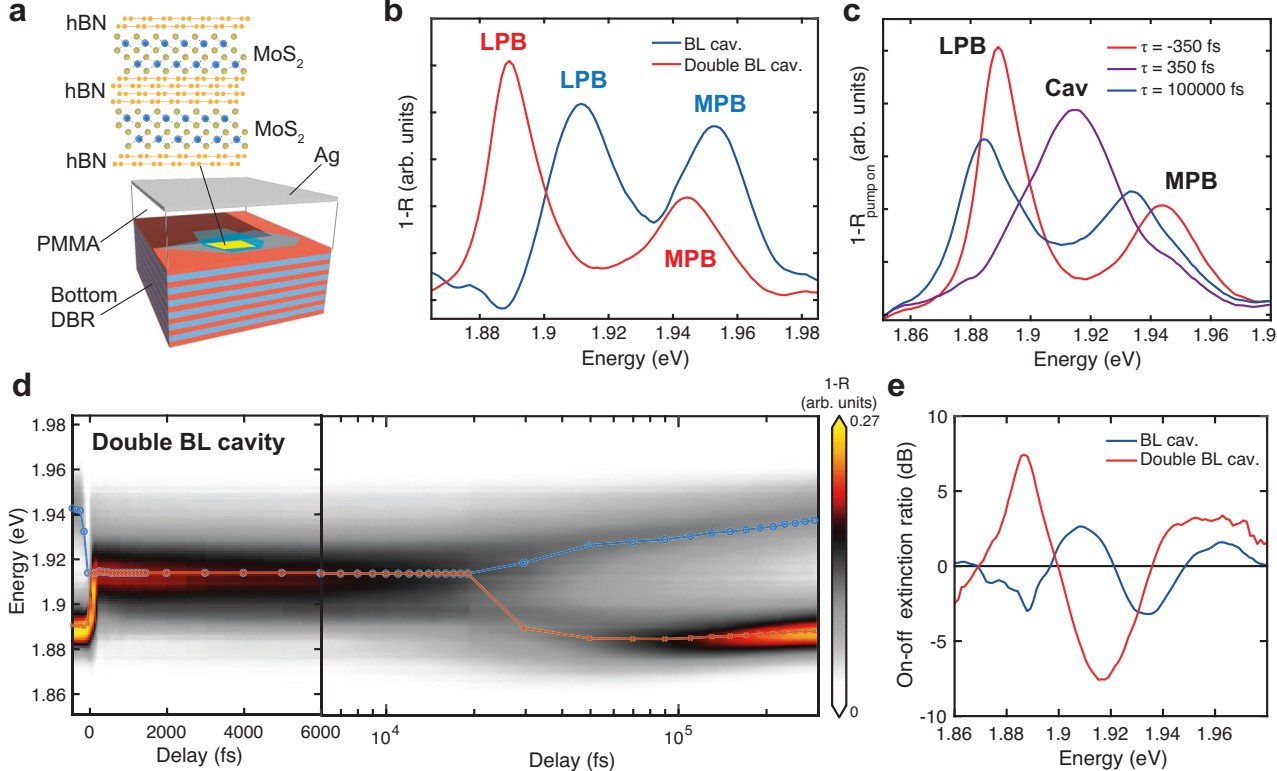

**Fig. 5 | Ultrafast switching of a double BL microcavity. a** Sketch of the microcavity embedding a double BL of MoS$_2$. **b** Static 1-R spectra of the single BL compared to the double BL microcavity, showing a redshift of the MPB-LPB and an increased polariton splitting in the latter. **c** 1-R spectra of the double BL microcavity excited with pump pulses at 1.91 eV and 8.25 pJ, taken at different pump-probe delays. **d** Color map of the 1-R spectra of the double BL microcavity versus pump-probe delay showing the ultrafast collapse and later revival of the SC. The orange (blue) line in overlay displays the fitted peak energy of the LPB (MPB) or the weakly coupled cavity mode. **e** Effective on/off extinction ratio calculated from the 1-R spectra taken at −350 fs and 350 fs, for the single (blue line) and double BL cavity (red line).

transfer rate. Furthermore, thermalization through exciton-phonon scattering is another important mechanism converting the bright states into momentum-dark states which may also gives smaller transfer rate upon hIX pumping. These scattering effects yield a faster recovery of SC, in agreement with our experimental observation (see Supplementary Note S8).

Another effect leading to faster recovery in the hIX pumping scheme is the mitigation of the Pauli blockade. In contrast to $X_{A-BL}$ pumping, the pumped exciton only shares holes with the probed exciton but not the electrons[54], see Fig. 4a, leading to a weaker saturation effect. This can also result in a faster recovery of SC if similar conditions as in $X_{A-BL}$ case are used (except the pump photon energy). Combining the effects of a smaller bright-to-dark exciton transfer rate and weaker Pauli blockade, the influence of the reservoir long-lived states is less significant exciting the hIX. Therefore, a faster recovery time of SC is easier to achieve in this case. Such fast recovery would allow to further increase the switching frequency, up to ≈1 THz. We underline that to achieve such hIX-induced Rabi quenching we use an excitation energy of 4.37 pJ, only moderately higher compared to the resonant excitation case. Higher pulse energies will increase the recovery time and the weak coupling time window.

## SC switching in a double BL microcavity

Finally, we fabricated a device comprising two vertically stacked bilayer MoS$_2$ separated by an hBN spacer of 40 nm. We placed this structure in a microcavity made of the same DBR and silver mirror used in the single BL cavity, with a PMMA spacer between the TMD stack and the top mirror (Fig. 5a). Similarly to microcavities with multiple

quantum wells[79], the SC is enhanced in this sample, as the Rabi splitting is increased to 55 meV due to the additional BL unit (see Supplementary Note S9 for the coupled oscillators model fit of the strongly coupled cavity dispersion). Figure 5b shows a comparison between the static 1-R spectra of a single BL (blue line) and a double BL (red line) cavity, taken at small angles. It clearly appears that the MPB-LPB peaks are more separated in the double BL cavity compared to the single BL sample, being also redshifted, as an effect of the more negative detuning of the former. The negative detuning also leads to a decrease of the LPB linewidth in the double BL cavity, being more cavity-like.

By exciting the double BL cavity with pump pulses resonant with $X_{A-BL}$, we observed again an ultrafast collapse of the MPB-LPB polariton peaks into a weakly coupled cavity mode, followed by their later recovery (Fig. 5c). We note that the pulse energy used for this experiment (8.25 pJ) was not adjusted to obtain a sub-ps SC recovery, but just to demonstrate the SC switching. Figure 5d shows the full dynamics of the SC collapse and recovery in the double BL cavity, where the energy separation between MPB and LPB (blue and red lines respectively) drops from ≈ 55 meV to zero immediately after the pump pulse. Considering the unperturbed SC condition as the on state of the optical switch and the weak coupling as the off state, we calculated the spectral power extinction ratio (ER) from the $1 − R$ spectra before and after the pump pulse, taken at − 350 fs and 350 fs respectively, where $ER(dB) = 10 \log((1 − R_{on})/(1 − R_{off}))$. The ER is increased significantly in the double BL cavity compared to the single BL device, over a broad energy range, as shown in Fig. 5e. For both the devices, the maximum ER in absolute value is reached around the energy of the weakly coupled cavity mode, between the LPB and MPB, which transmits (reflects) more during the off (on) state. While for the single BL the ER absolute

value reaches 3.2 dB at 1.932 eV, it is enhanced up to 7.5 dB at 1.915 eV in the double BL. It is worth to mention that the ER is also high in the spectral regions of the LPB and MPB, where it shows opposite sign, meaning that the optical switch can be used in direct or reverse mode just by changing the operational wavelength. We also tested the SC switching in a single BL cavity with a slightly negative detuning, similar to that of the double BL cavity, as shown in Supplementary Note S12. The negative detuning in this sample leads to a maximum ER of ~ 7 dB (Fig. S13c), higher compared to that of the single BL cavity shown in Fig. 5. Such value is similar to the maximum observed in the double BL sample, pointing out the importance of the detuning for a high contrast. However, in a single BL device the ER is maximized only in a narrow spectral region around 1.93 eV. A double BL cavity instead ensures maximum contrast in a broader energy range owing to the increased Rabi splitting, and consequently the larger energy separation between LPB and the cavity mode.

We note that working at high frequencies, the effective extinction ratio for a second switching event decreases due to the residual exciton population after the first pulse excitation (e.g. by about four times in the double pump pulse experiment shown in Fig. 3). We foresee that this drawback can be mitigated by reducing the long exciton decay component, for example suppressing the exciton scattering to dark states and using a different optical resonator with higher Q factor and smaller mode volume, to induce a strong Purcell effect.

## Discussion

In summary, we exploit the transient behavior of $MoS_2$ exciton-polaritons to demonstrate ultrafast optical switches using low pulse energies ( < 4 pJ), whose operational principle is based on the instantaneous transition from the strong to the weak coupling regime due to optical saturation of excitons, which can recover on the sub-picosecond timescale. The $MoS_2$ BL system uniquely combines nanometric thickness, large nonlinearities and large Rabi splitting with short lifetimes, with the latter enabling observation of the exceptionally fast recovery. SC switching can be performed in this platform even at ambient conditions, still using very low pulse energies, below 2 pJ. Furthermore, we show that by increasing slightly the pump pulse energies above the threshold for the SC collapse, the weak coupling time window can be significantly extended and deterministically controlled, being sensitive to energy variations of hundreds of femtojoules and below, crucially important for sensing and low light applications[80]. Such strongly fluence-dependent switching dynamics can be also exploited to emulate spiking neurons in novel neuromorphic computing architectures[81]. This system, also, offers additional degrees of freedom. It can operate at different excitation energies, for example in resonance with either intra- or interlayer excitons, leveraging on the strong interspecies interactions between these excitonic species, unique to the $MoS_2$ BLs. We foresee this property to be particularly useful for multiplexed logic operations[82]. Owing to the fast recovery of SC, we were able to perform subsequent switching events delayed by 4 ps, demonstrating an operational frequency of ≈250 GHz. Considering the sub-ps SC switching time, this frequency can be pushed up to 1 THz, surpassing even the fastest electronic transistors demonstrated so far[83]. We also demonstrate that using a microcavity with two stacked $MoS_2$ BLs can boost the Rabi splitting and greatly enhance the on/off extinction ratio, reaching a maximum of 7.5 dB in a single switching event. Compared to BEC-based polariton switching, our system does not need a high Q factor to obtain the switching effect. A further improvement of the optical resonator Q factor and a shorter exciton lifetime will lead to even greater on/off contrast for high frequency switching. For example, increasing the Q factor even by one order of magnitude will still result in a polariton lifetime below 1 ps, resulting at the same time in a polariton linewidth of few meV, hence obtaining a much higher on/off contrast.

Our work highlights TMD bilayers as a flexible system with rich physics in which sub-ps all-optical switching can be achieved and finely controlled. Such platform shows clear advantages compared to other materials in SC regime or even to TMD monolayers (see Supplementary Note S13 for a detailed comparisons with other systems).

The insights provided can be pivotal for the development of TMD-based high speed all-optical circuits. Moreover, the developed ultrafast nonlinear switching unit can improve the performance of optical neural networks[84,85] acting as an all-optical nonlinear activation function. Considering also the nanometric thickness of each hBN/BL/hBN stack, a microcavity could be filled by many TMD units, greatly increasing the Rabi splitting and subsequently the spectral shifts when used as ultrafast switches, which will lead to enhanced on/off extinction ratio. Moreover, the integration of electrical contacts in the microcavity structures[86] would enable the fabrication of electro-optical interfaces by tuning the electrostatic doping and electric field, which can provide giant shifts of the hIX energy in $MoS_2$ BLs[56], also enhancing their nonlinear interactions. Our SC switching approach can be extended also to other types of TMD homobilayers or even to moiré heterobilayers. In the latter case, the exciton confinement within the moiré potential will foster polariton nonlinear interactions[49], leading to optical saturation and SC quenching at lower exciton-polariton densities. Owing to the low pulse energies used, we observed no degradation of the devices after several switching experiments, even under ambient conditions, ensuring good long-term switching stability. Identifying strategies to suppress the slow exciton decay component will also ensure a high on/off extinction ratio for multiple switching events while working at very high frequencies. Optimizing the coupling of the TMD with a different optical resonator, e.g. waveguide resonances or nanophotonic structures, will enable the on-chip integration of multiple switching nodes within in-plane optical networks. Very small mode volumes and strongly localized light fields typical of such structures will also decrease the pulse energies required for the switching. Nanophotonic devices embedding TMD MLs that host quasi-bound states in the continuum modes with high Q factor have been recently demonstrated[87,88]. In such systems, the reflectivity (transmissivity) in the spectral regions around the uncoupled exciton energies can be moderately low (high). Hence, increasing the Q-factor will improve the on/off contrast by reducing the polariton linewidth, while still ensuring optical access to the excitons.

Developing ultrafast all-optical switches based on the transition from the strong to the weak coupling regime would be crucial also to unveil more exotic physical phenomena. The transition between the strong and weak coupling regime is linked to the observation of exceptional points, where the eigenvalues and eigenfunctions of the coupled systems coalesce, enfolding exotic physics arising from the non-Hermitian Hamiltonian describing such condition[89,90]. The encirclement of exceptional points in microcavities, controlling the detuning and the coupling strength, has been recently demonstrated[91,92], paving the way for the investigation of non-Hermitian physical phenomena, such as anomalous topological phases[93] and dissipative phase transitions[94,95]. Our platform offers a new approach to tune the system parameters for encircling the exceptional point on ultrafast timescales.

## Methods
### Sample fabrication
The hBN/MoS$_2$/hBN heterostructures were assembled using a poly-dimethylsiloxane (PDMS) polymer stamp method. The PMMA spacer for the monolithic cavity was deposited using a spin-coating technique, while a silver mirror of 45 nm thickness was thermally evaporated on top of it.

### Optical measurements
For the transient reflectivity measurements 100-fs pulses from an amplified Ti:Sapphire laser at 2 kHz repetition rate are used. The laser

output is split in two beams. A portion of the laser output is utilized to drive a non-collinear optical parametric amplifier (NOPA), which allows tuning the pump wavelength. The rest is used for the generation of the broadband white light probe pulse by focusing the beam on a sapphire plate. The delay between pump and probe pulses is controlled by a mechanical delay line. The pulses are combined collinearly and focused on the sample using a 50x objective, resulting in a spot size of $\approx 1.5\,\mu$m. The sample is kept in a helium cryostat at 8 K. The differential reflectivity ($\Delta R/R$) spectra are recorded at various time delays $\tau$ to track the changes induced by the pump. Specifically, the reflectivity spectrum of the probe with the pump on, $R_{\mathrm{PumpOn}}$, is compared at each delay with a reference spectrum obtained when the pump is off, $R_{\mathrm{PumpOff}}$. These are used to calculate $\frac{\Delta R}{R} = \frac{R_{\mathrm{PumpOn}} - R_{\mathrm{PumpOff}}}{R_{\mathrm{PumpOff}}}$, shown in the TR maps. The pump is orthogonally polarized with respect to the probe and it is filtered out by using a polarizer in the detection path. To remove any residual pump signal we also subtract a background spectrum taken without the probe to all the differential reflectivity spectra. The dynamic probe maps are extracted from a combination of the measured RC spectrum with the pump off and the $\Delta R/R$ maps, as $R_{\mathrm{PumpOn}} = R_{\mathrm{PumpOff}}(1 + \frac{\Delta R}{R})$. For the double pump pulses experiments, we use a thick YVO$_4$ birefringent crystal with optical axis rotated at 45° with respect to the vertical pump polarization, which produces a replica of the pulse with horizontal polarization delayed by $\approx 4$ ps. The rotation of a subsequent polarizer is changed to finely adjust the energy of each pulse in order to ensure the SC recovery after each excitation pulse.

## Transfer matrix method analysis

In order to extract the spectral and temporal evolution of the excitonic optical properties from the $\Delta R/R$ maps, we follow a procedure recently reported in refs. [58,96]. The transient reflectivity $R(\omega, \tau)$ of MoS$_2$ BL at each delay time is determined from the equilibrium reflectivity $R(\omega)$, which is reconstructed from the TMM fit of the static RC spectrum, and the transient reflectivity $\frac{\Delta R}{R}(\omega, \tau)$, following this relation:

$$R(\omega, \tau) = R(\omega)\left(\frac{\Delta R}{R}(\omega, \tau) + 1\right) \qquad (1)$$

Then, the dynamic RC is obtained by applying the formula: RC($\omega$, $\tau$) = $1 - (R(\omega, \tau)/R_{\mathrm{sub}})$, where the substrate reflectivity $R_{\mathrm{sub}}$ is simulated with the TMM. See Supplementary Note S7 for more details on the TMM simulations.

## Theoretical model for the polariton dynamics

To gain further insight into the dynamics in the system, we develop a mean-field model that captures the main trends in our experiment over different excitation regimes. The Hamiltonian corresponds to the coupled cavity-photon system, where the X$_{A-BL}$ mode (being the probed A exciton of a homobilayer) hybridizes with the cavity mode. The Hamiltonian reads

$$H = \begin{bmatrix} E_c + i\kappa & \frac{1}{2}g(n_X)\Omega_{A_{BL}} \\ \frac{1}{2}g(n_X)\Omega_{A_{BL}} & E_{A_{BL}} + i\gamma \end{bmatrix}, \qquad (2)$$

where $E_c$ and $\kappa$ are the energy and linewidth of the cavity photon, and $E_{A_{BL}}$ and $\gamma$ are the energy and linewidth of the probed exciton. In the Hamiltonian above $\Omega_{A_{BL}}$ is the Rabi splitting at weak pumping, and $g(n_X) = e^{-\alpha n_X}$ is the dimensionless nonlinear coupling, with $\alpha$ being the nonlinear phase space filling (saturation) coefficient[97]. The magnitude of the Rabi splitting is dependent on the total number of excitons in the system, $n_X$. In general $n_X(t) = n_p(t) + n_R(t)$ is time-dependent, and includes excitons (electron-hole pairs) from different states. Specifically, we separate the two fractions corresponding to $n_p$ and $n_R$ being the population of the pumped exciton and the long-lived excitons in

the reservoir. Crucially, both contribute to the nonlinear saturation effect. The dynamics of excitonic fractions can be described by rate equations defining the transfer and population redistribution, which read

$$\frac{dn_p}{dt} = -\gamma_p n_p - r n_p + \Theta(t), \qquad (3)$$

$$\frac{dn_R}{dt} = -\gamma_R n_R + r n_p, \qquad (4)$$

where $\gamma_p$ is the pumped exciton decay rate, $\gamma_R$ is the decay rate of the long-lived exciton in the reservoir, and $r$ is the rate constant for transferring the pumped excitons into the reservoir. The function $\Theta(t)$ depends on the pump laser profile in time. For example, $\Theta(t)$ can be a Heaviside step function to model the pump as an on-off switching field. In this model, we assume $\gamma_R \approx \gamma_p/100$ for the long-lived states corresponding to 100 ps decay time, and consider the decay timescale being similar to that of spin-forbidden dark states[98]. In fact, the decay time may be different, but this does not change our later conclusion in a qualitative way. Furthermore, when considering a spin-conserving process, we let the transfer rate be comparable to the pumped exciton decay rate[99], $r \approx \gamma_p$, such that this allows the pumped exciton transfer into the reservoir. With this, we find a good qualitative agreement between the theoretical spectrum and the experimental measurement (see Supplementary Note S8). Particularly, the theory demonstrated the excitation pulse energy dependence of the recovery time of SC.

## Inclusion and ethics statement

All collaborators of this study that have fulfilled the criteria for authorship required by Nature Portfolio journals have been included as authors, as their participation was essential for the design and implementation of the study. Roles and responsibilities were agreed among collaborators ahead of the research. This work includes findings that are locally relevant, which have been determined in collaboration with local partners. This research was not severely restricted or prohibited in the setting of the researchers, and does not result in stigmatization, incrimination, discrimination or personal risk to participants. Local and regional research relevant to our study was taken into account in citations.

## Data availability

The data generated in this study are available on Zenodo public repository, with the https://doi.org/10.5281/zenodo.15716409 (2025).

## Code availability

The computer codes and algorithms used to process the data included in this study are available from the corresponding authors upon request.

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

## Acknowledgements

CT acknowledges the European Union's Horizon Europe research and innovation program under the Marie Skłodowska-Curie PIONEER HOR-IZON-MSCA-2021-PF-GF grant agreement No 101066108. AG, CT, SDC and GC acknowledge funding from the European Horizon EIC Pathfinder Open program under grant agreement no. 101130384 (QUONDEN-SATE). This work reflects only authors' view and the European Commission is not responsible for any use that may be made of the information it contains. AIT and SR acknowledge financial support of the EPSRC grants EP/V006975/1, EP/V026496/1 and EP/S030751/1. OK and KWS acknowledge the support from UK EPSRC grant EP/X017222/1. DGL thanks the EPSRC for support via Programme Grant 'Hybrid Polaritonics' EP/M025330/1.

## Author contributions

AG, CL, CC and CT carried out the optical spectroscopy experiments with contribution from GDB and GS. CL and SR fabricated the 2D samples. KW and TT synthesized the high quality hBN. KG, PC, RJ and DGL fabricated the microcavities. AG, CL, CC, GDB and GS analyzed the data with contribution from SDC, AIT and GC. AG performed the transfer matrix simulations. KWS and OK developed the theory on dynamic optical saturation in microcavities. AG, CL and GC wrote the manuscript with contribution from all other co-authors. DGL, OK, SDC, AIT and GC managed various aspects of the project. SDC, AIT and GC supervised the project.

## Competing interests

The authors declare no competing interests.

## Additional information

[1]Dipartimento di Fisica, Politecnico di Milano, Piazza Leonardo Da Vinci 32, 20133 Milano, Italy. [2]NanoPhotonics Centre, Cavendish Laboratory, Department of Physics, JJ Thompson Ave, University of Cambridge, Cambridge, UK. [3]Department of Physics, University of Exeter, Stocker Road, EX4 4PY Exeter, UK. [4]Department of Physics, Xiamen University Malaysia, 49300 Sepang, Malaysia. [5]Department of Mechanical Engineering, Columbia University, New York 10027 NY, USA. [6]Dipartimento di Scienze Matematiche, Fisiche e Informatiche, Università di Parma, Parco Area delle Scienze 7/A, 43124 Parma, Italy. [7]School of Mathematical and Physical Sciences, University of Sheffield, Sheffield S10 2TN, UK. [8]Department of Physics, University of Cyprus, 1 Panepistimiou Avenue, 2109 Aglantzia, Nicosia, Cyprus. [9]Advanced Materials Laboratory, National Institute for Materials Science, 1-1 Namiki, Tsukuba 305-0044, Japan. [10]CNR-IFN, Piazza Leonardo da Vinci 32, Milano 20133, Italy. [11]These authors contributed equally: Armando Genco, Charalambos Louca.
✉e-mail: a.tartakovskii@sheffield.ac.uk; giulio.cerullo@polimi.it

