## [Transparent Peer Review file · Nature Communications]

Femtosecond switching of strong light-matter interactions in microcavities with two-dimensional semiconductors

Corresponding Author: Dr Giulio Cerullo

Version 0:

Reviewer comments:

Reviewer #1

(Remarks to the Author)

In this work the authors reported an ultrafast switching based on 2D TMD microcavity devices, using DBR-TMD-Silver mirror hybrid structure and pump-probe spectroscopy. I happened to read the previous version submitted to another Nature family journal. I conclude that the current submission has improved, with critical concerns being addressed properly. Therefore I would like to endorse my support to this work to be published in Nature communications.

Reviewer #2

(Remarks to the Author)

The manuscript by Genco et al. reports optical switching of strong light-matter coupling in bilayer MoS₂ microcavities. Ultrafast reflectivity measurements reveal a collapse of the Rabi splitting on sub-ps timescales and a full recovery of the Rabi splitting on a 100 ps timescale, using excitation powers of below 4 pJ per pulse. The authors demonstrate a repeated switching at pump-pair delays of 4 ps. The effect is reproduced at both cryogenic and room temperature. Such reversible strong-to-weak coupling on ultrafast timescales and low powers represents a meaningful step toward practical all-optical switching devices. While modulation or partial quenching of Rabi splitting in TMDC monolayers has been shown previously, this is, to my knowledge, among the first clear demonstrations of a complete and reversible strong-to-weak coupling transition in a 2D material system, especially using comparable low excitation powers.

After reading the previous referee reports and the responses by the authors, I find that the main concerns like switching contrast mechanisms, the choice of bilayers over monolayers, and quantitative polariton-density estimates have been thoroughly and convincingly addressed. Personally, I find the manuscript to be technically sound, clearly presented, and a step forward in the field beyond previous work on exciton nonlinearities in TMDCs. Additionally, detailed supplementary information that includes additional experimental data and theoretical modeling supports the conclusions substantially. A brief remark on the generalizability to other TMDC bilayers and on challenges such as long-term switching stability and device-level integration would broaden the outlook, but this is optional. Overall, as a result of the previous reviews and revisions, the study is ready for publication in Nature Communications.

Reviewer #3

(Remarks to the Author)

The authors report ultrafast switching between the strong and light-matter coupling in monolayer and bilayer MoS₂. The experimental results are interesting, showing strong and convincing effects. The manuscript is easy to follow and can trigger future progress in this field. However, I have a couple of comments, questions and concerns that should be addressed (see below). If the authors resolve these issues in the revised version of the manuscript, I can recommend publication of the paper in Nature Communications:

-My main concern refers to the definition of strong and weak coupling. The authors claim that the transition occurs when the Rabi splitting is equal to the unperturbed exciton linewidth (line 213), which is not correct. Rabi splitting between two

resonances is only observed (real valued) in the strong coupling regime. Strong coupling has the following criteria: a) the energy exchange rate (coherent coupling term) exceeds all loss rates, and b) the energy exchange between these resonances is larger than difference between the loss rates (see, for example, [Eur. J. Phys. 37 025802 (2016)]. Importantly, the transition is not sharp, as also seen for magnon polaritons [Physical Review Research 5, L012039 (2023)], and is dominated by exceptional points arising from the non-Hermiticity in the system [Eur. J. Phys. 37 025802 (2016)]. This was also experimentally exploited for 2D materials [Nat. Commun. 14, 1026 (2023)]. As the switching between these regimes is central to this work, I would suggest the authors to include a more rigorous definition of strong and weak coupling regimes and accordingly redefine their switching point.

-I would suggest the authors to amend Figure 3c based on the previous comment. Multi-peak fitting should allow the authors to include points below the $\Delta E_{\text{Pol}}/\Delta E_{\text{Pol Max}} \sim 0.45$ line, including error bars.

-The time-dependent behavior shown in Fig. 3c is interesting and very central to the work. While the authors ascribe those recovery steps to the population decay dynamics shown in Fig 1e (line 247), they provide only qualitative and theoretical explanations in the following paragraph. The authors reference Supplementary Note S6, but I cannot see how it reproduces the observed behavior of multiple recovery steps. For a more in-depth explanation, I suggest to fit the temporal dynamics of inter- and intralayer excitons as seen in Figure 1c,e, and compare the fitted rise and decay times directly to the decay and rise times observed in Fig. 3c. Extracting also energies and linewidths from Fig. 1c might help.

-When explaining the TR spectra in Fig. 1c,e, the authors miss a few contributions, including the optical Stark effect (lines 152-154) [Nat. Commun. 10, 5539 (2019), Nat Commun. 12, 4530 (2021)]. Generally, I miss more discussions on existing work on ultrafast exciton dynamics in TMDs, which is fundamental to presented observations.

-Fig. 2e shows a much slower recovery for the lower polariton branch compared to the middle polariton branch. I would suggest the authors to provide more explanations on this phenomenon. Introduction of Hopfield coefficients could help.

Reply letter

Reviewer #1 (Remarks to the Author):

In this work the authors reported an ultrafast switching based on 2D TMD microcavity devices, using DBR-TMD-Silver mirror hybrid structure and pump-probe spectroscopy. I happened to read the previous version submitted to another Nature family journal. I conclude that the current submission has improved, with critical concerns being addressed properly. Therefore I would like to endorse my support to this work to be published in Nature communications.

We thank the Reviewer for the very positive remarks about our work and for recommending its publication.

Reviewer #2 (Remarks to the Author):

The manuscript by Genco et al. reports optical switching of strong light-matter coupling in bilayer MoS₂ microcavities. Ultrafast reflectivity measurements reveal a collapse of the Rabi splitting on sub-ps timescales and a full recovery of the Rabi splitting on a 100 ps timescale, using excitation powers of below 4 pJ per pulse. The authors demonstrate a repeated switching at pump-pair delays of 4 ps. The effect is reproduced at both cryogenic and room temperature. Such reversible strong-to-weak coupling on ultrafast timescales and low powers represents a meaningful step toward practical all-optical switching devices. While modulation or partial quenching of Rabi splitting in TMDC monolayers has been shown previously, this is, to my knowledge, among the first clear demonstrations of a complete and reversible strong-to-weak coupling transition in a 2D material system, especially using comparable low excitation powers. After reading the previous referee reports and the responses by the authors, I find that the main concerns like switching contrast mechanisms, the choice of bilayers over monolayers, and quantitative polariton-density estimates have been thoroughly and convincingly addressed. Personally, I find the manuscript to be technically sound, clearly presented, and a step forward in the field beyond previous work on exciton nonlinearities in TMDCs. Additionally, detailed supplementary information that includes additional experimental data and theoretical modeling supports the conclusions substantially. A brief remark on the generalizability to other TMDC

bilayers and on challenges such as long-term switching stability and device-level integration would broaden the outlook, but this is optional. Overall, as a result of the previous reviews and revisions, the study is ready for publication in Nature Communications.

We appreciate the Reviewer's positive feedback on our work and their recommendation for its publication.

Actions:

We added the following sentences about the future outlooks of our approach to the Discussion section of our paper:

“Our SC switching approach can be extended also to other types of TMD homobilayers or even to moire’ heterobilayers. In the latter case, the exciton confinement within the moire’ potential will foster polariton nonlinear interactions [Zhang et al. Nature 591.7848 (2021): 61-65], leading to optical saturation and SC quenching at lower exciton-polariton densities. Owing to the low pulse energies used, we observed no degradation of the devices after several switching experiments, even under ambient conditions, ensuring good long-term switching stability. Identifying strategies to suppress the slow exciton decay component will also ensure a high on/off extinction ratio for multiple switching events while working at very high frequencies. Optimizing the coupling of the TMD with a different optical resonator, e.g. waveguide resonances or nanophotonic structures, will enable the on-chip integration of multiple switching nodes within in-plane optical networks.”

Reviewer #3 (Remarks to the Author):

The authors report ultrafast switching between the strong and light-matter coupling in monolayer and bilayer MoS₂. The experimental results are interesting, showing strong and convincing effects. The manuscript is easy to follow and can trigger future progress in this field. However, I have a couple of comments, questions and concerns that should be addressed (see below). If the authors resolve these issues in the revised version of the manuscript, I can recommend publication of the paper in Nature Communications:

We are glad that the Reviewer appreciated our work and recommended its publication. Below we provide a point-by-point answer to the raised issues.

-My main concern refers to the definition of strong and weak coupling. The authors claim that the transition occurs when the Rabi splitting is equal to the unperturbed exciton linewidth (line 213), which is not correct. Rabi splitting between two resonances is only observed (real valued) in the strong coupling regime. Strong coupling has the following criteria: a) the energy exchange rate (coherent coupling term) exceeds all loss rates, and b) the energy exchange between these resonances is larger than difference between the loss rates (see, for example, [Eur. J. Phys. 37 025802 (2016)]. Importantly, the transition is not sharp, as also seen for magnon polaritons [Physical Review Research 5, L012039 (2023)], and is dominated by exceptional points arising

from the non-Hermiticity in the system [Eur. J. Phys. 37 025802 (2016)]. This was also experimentally exploited for 2D materials [Nat. Commun. 14, 1026 (2023)]. As the switching between these regimes is central to this work, I would suggest the authors to include a more rigorous definition of strong and weak coupling regimes and accordingly redefine their switching point.

We agree with the Reviewer on the fact that there are more rigorous definitions of Strong Coupling (SC) than the one we used, such as that proposed by the Reviewer, $\Omega_R \geq |\gamma_{\text{cav}} - \gamma_{\text{exc}}|$. Here we demonstrate that changing this definition does not change our results conceptually.

Considering the Reviewer's next comment, we used a double-peak fit for the transient reflectivity data of the cavity pumped with 3.75 pJ, even below the SC threshold that we originally set ($\Delta E_{\text{pol}} \geq \text{unperturbed } \gamma_{0 \text{ exc}}$). The results are shown in Fig. R1a-b, which clearly demonstrates that a reasonable fit cannot be obtained for points below such a threshold.

For an accurate definition of the SC threshold, it should be also considered that while the cavity linewidth (i.e. its losses) remains approximately the same in all the experiments (~ 10 meV), the exciton linewidth changes with fluence and time because of excitation-induced dephasing. In order to extract the transient intralayer effective exciton linewidth ($\gamma_{\text{exc}}(t)$), we performed a full lineshape analysis on the time-dependent reflectivity of the out-of-cavity sample, measured with a pump fluence comparable to the one used for the switching experiments on the cavity (see reply to question n.3 for more details).

Fig. R1b compares the polariton splitting (ΔE_{pol}) with the values of $\gamma_{0 \text{ exc}}$ and $|\gamma_{\text{cav}} - \gamma_{\text{exc}}(t)|$ at each delay time. We point out that, to obtain the exact value of $\Omega_R(t)$, the cavity dispersion should be measured at each delay and fitted with a coupled oscillator model, but this is technically unfeasible for us. Hence, knowing that the cavity is close to zero-detuning for X_A , we approximately consider $\Omega_R(t) \approx \Delta E_{\text{pol}}(t)$. According to Fig. R1b, even taking into account the additional broadening due to excitation-induced dephasing, the SC switching still occurs on a sub-picosecond time scale, between ~ 50 fs and ~ 700 fs. Using the dynamic exciton linewidth to set the strong-to-weak coupling threshold would actually lower the pulse energy required for a complete SC switching. Setting the SC threshold at $\Delta E_{\text{pol}} \geq \gamma_{0 \text{ exc}}$ is instead more conservative.

Figure R1: a) Results of Gaussian fits of the cavity transient reflectivity extracted from Fig. 2d of the main text, fitting all the spectra with two peaks. The blue (orange) trace refers to the MPB (LPB) peak energy, while the vertical bars depict the linewidth of the polariton modes (Full Width Half Maximum, FWHM). b) Polariton energy difference (blue dots) extracted from panel a) as a function of delay time. The error bars depict the confidence intervals of the fits. The black dashed curve shows the difference $|\gamma_{\text{cav}} - \gamma_{\text{exc}}(t)|$, while the red dashed-dotted line refers to the unperturbed exciton linewidth, $\gamma_{0 \text{ exc}}$.

-I would suggest the authors to amend Figure 3c based on the previous comment. Multi-peak fitting should allow the authors to include points below the $\Delta E_{\text{pol}}/\Delta E_{\text{pol}} \text{ Max} \sim 0.45$ line, including error bars.

Actions:

Based on the previous reply, we modified Fig. 3c of the main text (shown below as Fig. R2) including the error bars from the fits and adding the trend of $|\gamma_{\text{cav}} - \gamma_{\text{exc}}(t)|$ together with the line related to $\gamma_{0 \text{ exc}}$. We did not include points for polariton splittings $\Delta E_{\text{pol}} < \gamma_{0 \text{ exc}}$ being not

meaningful because of the large uncertainty. We also blurred the yellow area depicting the weak coupling window.

Figure R2: MPB-LPB energy difference as a function of delay time for ultrafast SC switching experiments in BL cavities at different pump pulse energies, normalized to the value before excitation. Error bars are drawn from the confidence interval of the double peak fits. The dashed black curve shows the trend of the exciton-cavity linewidth difference, while the red dashed-dotted line shows the linewidth of the unperturbed exciton, both normalized by the polariton splitting value before excitation. The weak coupling time window duration can be tuned by changing the excitation pulse energy.

In addition, in the main text we changed the sentences below:

”We consider that the SC switching has been reached when the Rabi splitting is equal to the unperturbed exciton linewidth (the FWHM of X_{A-BL} is ~ 20 meV in static conditions). On the other hand, considering as switching threshold the splitting being equal to the broader polariton linewidths would be a less stringent criterion. In the latter case the pump energy to induce a peak splitting collapse would be even lower, also resulting in a faster SC recovery.”

with the following:

“To a first approximation, we can consider that the strong to weak coupling full transition is reached when the Rabi splitting is equal to the unperturbed exciton linewidth (the FWHM of X_{A-BL} , $\gamma_{0\text{ exc}}$, is ~ 20 meV in static conditions). A more precise definition of strong to weak coupling threshold implies that the energy exchange between cavity and exciton resonances is larger than the difference between the loss rates [Eur. J. Phys. 37 025802 (2016), Nat. Commun. 14, 1026 (2023)]. On the other hand, considering in our case the exciton line broadening caused by excitation-induced dephasing, this becomes a less stringent criterion, as discussed later in this section (Fig. 3c).”

We also added the paragraph below:

“The red dashed horizontal line in Fig.3c represents the threshold when the polariton splitting is smaller than the unperturbed exciton linewidth, while the black dashed curve shows the time-dependent normalized difference between exciton and cavity linewidths. While the cavity linewidth remains approximately the same in all the experiments (~10 meV), the exciton linewidth changes with fluence and time because of excitation-induced dephasing. We extract the transient X_{A-BL} linewidth, $\gamma_{exc}(t)$, analysing the time-dependent reflectivity of the out-of-cavity sample, excited with a pump fluence comparable to the ones used in the cavity experiments (see Supplementary Note S6). Using the exciton-cavity linewidth difference to set the threshold for SC, the switching is not as sharp, but it still occurs in a sub-picosecond time window, between ~50 fs and ~700 fs, pumping with 3.75 pJ. On the other hand, using such a definition for the strong to weak coupling transition, the pulse energy to induce the SC collapse will decrease.”

-The time-dependent behavior shown in Fig. 3c is interesting and very central to the work. While the authors ascribe those recovery steps to the population decay dynamics shown in Fig 1e (line 247), they provide only qualitative and theoretical explanations in the following paragraph. The authors reference Supplementary Note S6, but I cannot see how it reproduces the observed behavior of multiple recovery steps. For a more in-depth explanation, I suggest to fit the temporal dynamics of inter- and intralayer excitons as seen in Figure 1c,e, and compare the fitted rise and decay times directly to the decay and rise times observed in Fig. 3c. Extracting also energies and linewidths from Fig. 1c might help.

Actions:

Following the Reviewer’s suggestion, we compared quantitatively the temporal dynamics of the polariton splitting with the dynamic amplitudes of the intra and interlayer excitons in the out-of-cavity sample, observing a very good match. We added the related discussion in a new Supplementary Note, reported also below. As requested by the referee, we also extracted the energies and linewidths from Fig.1c, now presented in an additional Supplementary Note.

“Supplementary Note S6: Excitons versus polaritons dynamics in MoS₂ BL

In this section, we compare the temporal dynamics of the polariton splitting with the transient optical behaviour of the intra and interlayer excitons in the out-of-cavity sample, excited with a narrow-band pump in resonance with X_{A-BL} , using comparable pump fluences. For the cavity experiments shown in Fig. 2 of the main paper we used a fluence of 212 $\mu\text{J}/\text{cm}^2$, hence we excite the out-of-cavity sample with ~30 $\mu\text{J}/\text{cm}^2$ in order to obtain similar excitation densities on the TMD, considering a transmittance of the top silver mirror of about 15%. The transient RC pump-probe map measured in these conditions is shown in Fig. R3.

Using the same procedure based on the Transfer Matrix Method (TMM) that we present in the main text (see Methods), we fit the experimental RC spectra at each delay with Lorentzian functions to extract the dynamic lineshape of the different excitonic species. From such analysis

we can isolate the contribution of the time-dependent excitonic line broadening for intralayer excitons (X_A).

Figure R3: Dynamic RC versus pump-probe time delay and probe photon energy for the MoS₂ BL out-of-cavity sample, excited with $\sim 30 \mu\text{J}/\text{cm}^2$.

Figure R4: a) Normalized exciton peak amplitude variation of X_A and hIX (red and blue dots respectively), extracted from the dynamic RC fits at different time delays. The experimental dynamics are fitted with multi-exponential curves (solid lines) in order to extract rise and decay times. b) Normalized polariton splitting as a function of pump-probe delay (green dots) extracted from Fig. 2d of the main text, fitted with a multi-exponential curve (solid line) to extract rise and decay times.

Figure R4a shows the transient peak amplitude variation of X_A and hIX , which follows a double exponential decay. We fit the experimental exciton dynamics using a multi-exponential model, comprising a rise and two decay components, convoluted with the instrument response function of our setup [Trovatello et al, Nat Commun 11, 5277 (2020)]. We apply the same model to the transient polariton splittings (Fig. R4b), extracted from the data in Fig. 2c. In this case we fitted the cavity data with two peaks even below the SC threshold, despite the large uncertainties.

The table below summarizes the fitted values for the rise and decay times in the different cases, showing a very good match between the polariton splitting times and the X_A ones. Therefore, we conclude that SC collapse and recovery times are ruled by the optical saturation dynamics of the uncoupled excitons, or in other words by their formation and depopulation times.”

	τ_{rise} (fs)	τ_1 (fs)	τ_2 (ps)
X_A	60±30	610±40	47±1
hIX	105±20	1025±80	80±3
ΔE_{pol}	90±30	455±100	32±3

“Supplementary Note S2: Exciton energies and linewidths transient behaviour

In this section we present the time-dependent exciton linewidths (Fig. R5a) and line shifts (Fig. R5b) extracted from the pump-probe data taken on the MoS₂ BL excited with low fluence (Fig. 1c of the main text), using the fitting procedure used to obtain the transient exciton peak amplitudes shown in Fig. 1e.

Immediately after pump excitation, both the X_A and hIX lines broaden significantly due to excitation-induced dephasing. This effect fades out in two distinct time-scales following the decay of exciton population, the first faster decay occurring below 1 ps. We observe that the first decay for hIX is faster than for X_A , the former being not pumped directly but interacting only with the holes created in the valence band shared with X_A .

Regarding the exciton line shifts, at short delay times, while the immediate blueshift of hIX can be attributed to strong repulsive Coulomb interactions, the ultrafast blueshift of X_A can be related also to the optical Stark effect [Nat. Commun. 10, 5539 (2019), Nat Commun. 12, 4530 (2021)]. However, a detailed explanation of these effects in our samples goes beyond the scope of this work.

Figure R5: a) Transient linewidths Γ_{exc} of X_A (red line) and hIX (blue line), extracted from fitting the dynamic RC in Fig.1 of the main paper. b) X_A (red line) and hIX (blue line) transient peak shifts, extracted from fitting the dynamic RC in Fig.1 of the main paper.”

-When explaining the TR spectra in Fig. 1c,e, the authors miss a few contributions, including the optical Stark effect (lines 152-154) [Nat. Commun. 10, 5539 (2019), Nat Commun. 12, 4530 (2021)]. Generally, I miss more discussions on existing work on ultrafast exciton dynamics in TMDs, which is fundamental to presented observations.

Actions:

Hereafter we present a new paragraph that we included in the main text with more discussions on ultrafast exciton dynamics in TMDs. Other insights can be also found in the new Supplementary Note S2 (see previous reply).

“The ultrafast nonlinear optical response of mono and few-layers TMDs has been studied extensively in the past [Aivazian et al, 2D Materials 2017, 4, 025024; Singh et al, Phys. Rev. Lett. 2014, 112, 216804; Dal Conte et al, Trends In Chemistry 2020, 2, 28– 42]. Transient exciton line shifts in TMDs are usually ascribed to Coulomb interactions at short time scales (few ps) [Trovatello et al, Nano Lett. 2022, 22 (13), 5322–5329], or bandgap renormalization [Pogna et al, ACS Nano 2016, 10 (1), 1182–1188], and to transient heating effects [Ruppert et al, Nano Lett. 2017, 17, 2, 644–651] at longer times (from tens to hundreds of ps). Exciting TMD monolayers close or below the exciton energy leads also to strong and instantaneous (within the pump pulse duration) line shifts due to the optical Stark effect [Nat. Commun. 10, 5539 (2019), Nat Commun. 12, 4530 (2021)]. High exciton densities in TMDs lead to optical saturation, due to phase-space filling (i.e. Pauli blocking) [Ruppert et al, Nano Lett. 2017, 17, 2, 644–651], and line broadening caused by excitation-induced dephasing [Katsch et al, Phys. Rev. Lett. 2020, 124 (25), 257402]. Tracking the time-dependent exciton saturation in ultrafast pump-probe experiments allows monitoring the exciton population dynamics.

In MoS₂ BLs we observe a bi-exponential population decay with a fast and a slow component. While in MLs the fast decay is usually attributed to radiative and non-radiative relaxation processes of bright excitons [Poellmann et al, Nature Mater 14, 889–893 (2015)], in BLs it is more probably related to electron-phonon inter-valley scattering processes from the K points to the lowest energy point of the Brillouin zone [Nie et al, J. Phys. Chem. C 2015, 119 (35), 20698–20708; Din et al, 2D Mater. 2021, 8 (2), 025018; Nie et al, ACS Nano 2014, 8 (10), 10931–10940]. The slow decay component can be related to phonon-assisted recombination from dark states [Poellmann et al, Nature Mater 14, 889–893 (2015)] or defect-mediated non-radiative recombination [Wang et al, Nano Letters. 2015, 1, 339-345].”

-Fig. 2e shows a much slower recovery for the lower polariton branch compared to the middle polariton branch. I would suggest the authors to provide more explanations on this phenomenon. Introduction of Hopfield coefficients could help.

Actions:

We added the following sentences to the main text to explain the different recovery times of the polariton branches:

“The two polariton branches show different recovery times depending on their Hopfield coefficients, and in particular on their photonic component. In fact, a polariton branch with a larger photonic character will be closer in energy to the weakly coupled cavity mode, leading to a faster recovery. Therefore, a positive detuning benefits the MPB recovery over the LPB one, as shown in Fig. 2d, while the opposite happens for negative detunings (see Fig. S10).”